# Planktonic Copepoda along the Confinement Gradient of the Taranto Sea System (Southern Italy) after Lockdown of Human Activities Due to the COVID-19 Pandemic

Genuario Belmonte [1],*, Giuseppe Denti [2] and Fernando Rubino [2]

1   Department of Biological and Environmental Sciences and Technologies, University of Salento, 73100 Lecce, Italy
2   Unit of Taranto, CNR Water Research Institute (IRSA), 74123 Taranto, Italy; giuseppe.denti@irsa.cnr.it (G.D.); fernando.rubino@irsa.cnr.it (F.R.)
*   Correspondence: genuario.belmonte@unisalento.it

**Abstract:** This study was conducted in the sea system of Taranto (south Italy) to explore the consequences of the COVID-19 lockdown of all human activities on zooplankton abundance and composition. Copepoda were selected as the best indicators and four different dates were taken to represent the annual variability. The availability of samples from past collections (15 and 30 years ago) allowed comparison with previous situations. The Copepoda community in the most confined part of the system (stations MPI and MPII) was dominated by small-sized species and showed new arrivals, including *Acartia tonsa*, *Centropages hamatus,* and *Pseudodiaptomus* sp. The first inlet of Mar Piccolo (MPI) showed an unusually high number of species in the summer of 2020, just at the end of the lockdown period (March–May 2020). The evident growth of species richness at station MPI, and only during the summer of 2020, suggests a role of the lockdown period on the zooplankton composition. The high species richness in the post-lockdown period was probably the result of ceasing the disturbance caused by ship/boat traffic at the MPI site, which is heavily affected by daily human activities at sea.

**Keywords:** zooplankton; Copepoda; confined environment; COVID-19 lockdown; *Acartia tonsa*; *Centropages hamatus*; *Pseudodiapomus* sp.

## 1. Introduction

Zooplankton in confined coastal environments are typically subjected to more seasonal oscillations in abundance and species composition compared with those in neighboring open seas [1]. In general, enhanced water trophism (common in confined coastal systems) and climate warming are associated with driving the community structure towards the affirmation of small-sized species, a higher dominance, and smaller species richness [2]. Confined coastal environments, in addition, are particularly affected by human activities, both by the heavy presence of human settlements and the low renewal time of their waters (according to the definition of confinement by [3]). Such habitats, as a consequence of these environmental drivers, should be particularly sensitive to changes in abiotic and biotic parameters. What is interesting to establish, however, is if the variation (in terms of biomass and species composition) can be attributed to intrinsic properties of the community (internal dynamics) or to external changes, either natural (climate change) or artificial (directly derived by human presence). The dynamics internal to the communities in confined environments comprise the possibility for each species to disappear for long periods from the water column, without abandoning the site, thanks to the strategy of producing resting stages [4] that wait in the bottom sediments for the return of favorable conditions even after years. The detailed Zooplankton composition based on such a strategy is also unpredictable, because the success of the anticipated hatching and germination is affected by casualty, and "bet winners" (see the Bet Hedge strategy of [5]), are those species

that, year after year, wake up and re-start their life cycle using available energetic resources before the other species, and even before the full resource availability (see [6]).

The individuation of natural ecosystem dynamics is of pivotal importance in ascertaining human responsibilities in environment modification, and understanding if and how human-derived changes in the environment affect the return of the same species after a period of absence.

The Taranto sea system is an important site for such a study for many reasons: (a) the system has evident and different grades of confinement, (b) Taranto city (192,000 inhabitants) represents a high-impacting human settlement, (c) the ship/boat traffic is responsible for continuous NIS arrivals from other geographic areas, and (d) the existence of a scientific institution guarantees the availability of data (physical, chemical, and biological) from the past for comparison.

For these reasons, the inner part of the Taranto sea system—the Mar Piccolo—was selected as one of the study sites in the European LTER (Long Term Ecological Research) network.

The last human-derived impact on the marine-confined environment was identified in the lockdown consequent to the COVID-19 pandemic, which stopped activities (of all kinds) for two months in March–May 2020. In this case, stopping human activities was considered a possible source of variation in a normally disturbed system [7], and analysis of the planktonic Copepoda assemblage is proposed as an indicator of such a variation in the disturbance. Low values of Chl-*a* concentration were observed in both inland and marine North Adriatic waters during the 2020 lockdown, and statistics suggest an anthropogenic (positive) effect on the environment due to the lockdown restrictions [8].

The most recent studies of zooplankton from Taranto [1,9] (based on samples collected in 1990–1991 and 2005–2006) established the general trend of smaller, seasonal, and resting-stage-producing species in confined parts of the Taranto sea system.

The aim of the present study was to establish if and how the lockdown in the spring of 2020 produced variations in the composition and abundance of zooplankton in individual parts or the whole Taranto sea system. The samples collected in preceding studies, for this purpose, have been re-analyzed to enable data comparability.

## 2. Materials and Methods

### 2.1. Study Site

The Taranto sea system is composed of four delimited sea areas (Gulf of Taranto, Mar Grande, and the two basins of Mar Piccolo, namely 1st Inlet and 2nd Inlet) in southeastern Italy (Figure 1).

The Gulf of Taranto, Mar Grande, and Mar Piccolo are aligned along a progressive confinement gradient that also overlaps a eutrophic gradient [10,11]. Mar Piccolo is a semi-enclosed sea divided by a promontory into 2 basins; the 1st Inlet (MPI, maximum depth of 13 m) is directly connected with the MG through two narrow canals, and the 2nd Inlet (MPII, maximum depth of 9 m) represents the most confined part of the system. MP is characterized by limited water circulation, and in the absence of significant tidal excursions (annual maximum of 28 cm), the exchanges with MG depend entirely on two relatively narrow canals. Water salinity and temperature in the two MP basins are affected by the presence of 34 brackish water submarine springs [12], as well as by small surface water courses. The total flow of these inputs is seasonal, reaching a maximum of 0.01 $km^3$ $week^{-1}$ (the total water volume in MP has been estimated at 0.15 $km^3$), allowing Strusi and Pastore [13] to consider the system as corresponding to an estuary.

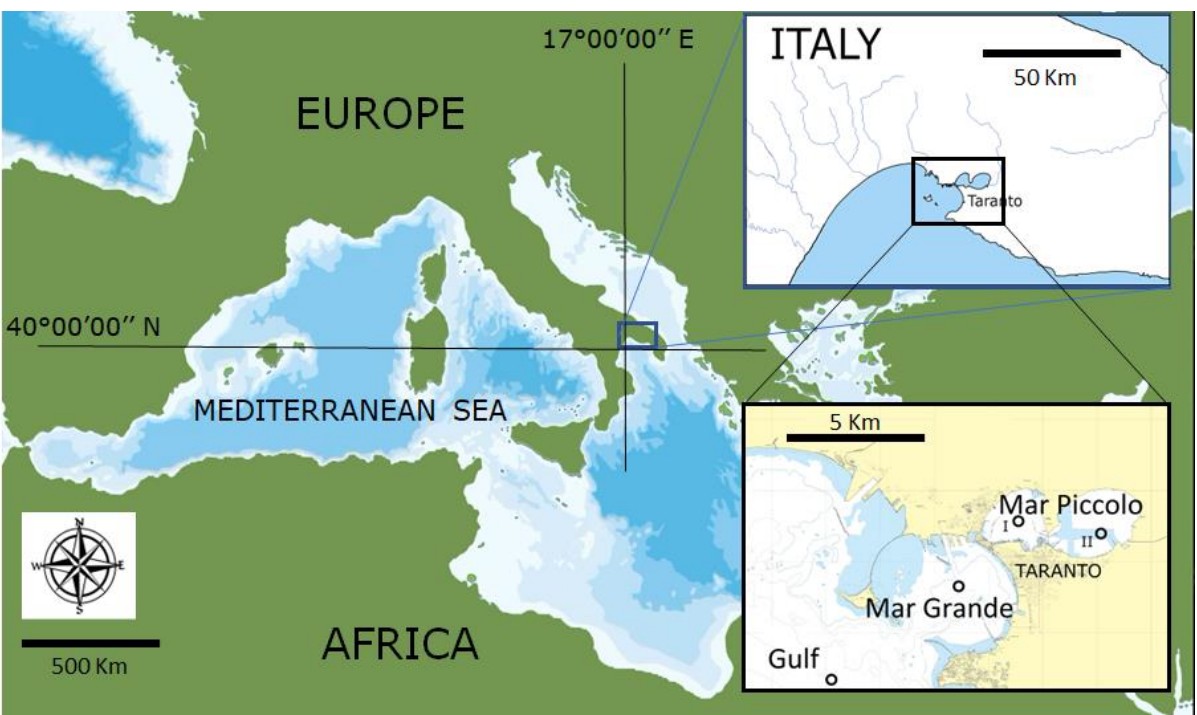

**Figure 1.** Study site. Black circles on the map of Taranto indicate the positions of the stations.

MG is a typical neritic area, but its primary production is high and similar to that of MP [12], probably as a consequence of human-derived organic contamination associated with the presence of the city of Taranto (about 192,000 inhabitants). MG has a maximum depth of 32 m and is separated from the open sea by the Cheradi islands and the breakwater outside the new harbor.

The open sea considered in the present study is the northern corner of the Gulf of Taranto (G), a square of about 13,700 km$^2$ with a maximum depth of about 2000 m, open on the South–East side to the northern Ionian Sea (central Mediterranean).

*2.2. Field Work*

Zooplankton samples were collected in each season for one year, in July and October of 2020, and in January and April of 2021, at four stations: G (Gulf), MG (Mar Grande), MPI (Mar Piccolo 1st inlet), and MPII (Mar Piccolo 2nd inlet), towing a plankton net with an 80-μm mesh size (mouth diameter of 56 cm) from the bottom to the surface along the water column. The water depths at the stations were 60 m, 25 m, 12 m, and 9 m, respectively.

At each station on each sampling date, two independent replicates were obtained using different tows. The present study focused on Copepoda because their number and species richness is representative of the whole zooplankton in coastal marine environments. The mesh size was chosen because specimens with small sizes (adults of small species and/or larvae and juveniles of large species) generally have an importance that increases with the confinement grade [1,2,14]. Collection was timed to start in summer, to allow the Copepoda born during lockdown (March–May 2020) to reach adulthood to enable their identification at the species level.

Samples were immediately fixed in ethanol (final concentration 75–85%) and transferred to the laboratory for qualitative/quantitative analysis.

Chemical–physical parameters (temperature, salinity, and pH) were measured using a multi-parametric probe (IDROMARAMBIENTE IP050D, Genoa, Italy) at each station and on each date.

The volume of filtered water for each towing was estimated using a flow meter (HYDRO—BIOS Model 438115) placed at the mouth of the plankton net.

*2.3. Laboratory Work*

The Copepoda in each sample were identified and enumerated. The identification was carried out up to the lowest taxonomic level (species) for adults. Nauplii and juveniles were classified by family. Adult specimens of the Calanoida species are conserved as part of the marine biodiversity collection at the Marine Biology Museum of the University of Salento.

Abundance data are reported as indiv. m$^{-3}$. The identification of species relied upon [15–17]. Species new to Taranto and/or Italian fauna were analyzed in detail and drawn using a camera lucida positioned on a Zeiss compound microscope (Axiovert S100) at 400× magnification.

*2.4. Comparison with Historical Collections*

The methodological approach for the 2020–2021 study (time 3, T3) was chosen to allow comparison with the Copepoda assemblages from the preceding studies conducted in 1990–1991 (T1) and 2005–2006 (T2), 30 and 15 years from the present study. The 1990–1991 samples [7] were limited to the family taxonomic level. The 2005–2006 samples [1] were more accurate and were collected using meshes of 2 sizes (80 and 200 μm) and 3 replicates per mesh size, at each station. For these reasons, samples from past collections (available at the Museum of Marine Biology of the University of Salento) were re-examined with the aim of homogenizing the data to allow comparisons among the three periods (T1–T3). In detail, the samples from 1990 to 1991 were re-analyzed to identify Copepoda at the species level to compare with the present situation. Regarding the data collected from 2005 to 2006 [1], 2 replicates from corresponding periods of the year (months) and 80-μm net sizes were isolated to obtain a data matrix comparable with the data in the present study.

*2.5. Statistical Analysis*

All univariate and multivariate analyses were performed using PRIMER v.6 (Primer-E Ltd., Plymouth, UK).

A data matrix (taxa vs. samples; 69 rows vs. 48 columns) was constructed from the average abundance values (ind. m$^{-3}$) calculated from the replicates collected at each sampling site in the three years of this study and used to calculate the Bray–Curtis similarity after 4th root transformation to normalize the data and downweigh the importance of the most abundant species.

The PRIMER 'DIVERSE' routine was used on untransformed data to calculate the taxonomic richness (S), taxon abundance (N), Pielou's evenness index J' [18], and Shannon–Weaver diversity index H'.

Bi-dimensional representations of the statistical comparisons of the samples collected at the four sites were obtained using non-parametric multidimensional scaling (nMDS).

The statistical significance of the spatial differences identified by nMDS was tested using one-way analysis of similarities (the PRIMER 'ANOSIM' routine) to determine whether replicates from the four investigated sites were more different than replicates within sites.

## 3. Results

Abiotic features of the environment confirmed a situation already known, with the external station (G) showing the highest differences between bottom and surface values, and the lowest annual variability of measured parameters. On the contrary, the innermost station (MPII) showed consistent parameters along the water column and the highest annual variability (Table 1). Comparisons between periods showed a tendency towards high average temperatures from T1 to T3 and a slight diminution of salinity. These differences, however, were not statistically significant.

**Table 1.** Environmental data. The volume of water filtered per sample is an average of two replicates. SUM, AUT, WIN, and SPR are the seasons (with the month indicated in brackets).

| | | 1990–1991 | | | | | 2005–2006 | | | | | 2020–2021 | | | | |
|---|---|---|---|---|---|---|---|---|---|---|---|---|---|---|---|---|
| | | SUM (Jul.) | AUT (Oct.) | WIN (Jan.) | SPR (Apr.) | Average 1990–1991 | SUM (Jul.) | AUT (Oct.) | WIN (Jan.) | SPR (Apr.) | Average 2005–2006 | SUM (Jul.) | AUT (Oct.) | WIN (Jan.) | SPR (Apr.) | Average 2020–2021 |
| **Station G**<br>Water column, 50 m | lat. 40°25′08″ N, long. 17°10′45″ E | | | | | | | | | | | | | | | |
| Sample vol., m$^3$ | | 3.3 | 3.5 | 3.6 | 3.4 | **3.45** | 4.40 | 4.10 | 4.70 | 4.30 | **4.38** | 3.90 | 4.30 | 4.50 | 4.00 | **4.18** |
| Temp. °C | surface | 25.10 | 21.10 | 14.20 | 14.80 | **18.80** | 25.20 | 26.10 | 14.80 | 17.00 | **20.78** | 26.10 | 26.60 | 14.20 | 13.10 | **20.00** |
| | bottom | 17.50 | 15.50 | 14.00 | 13.90 | **15.23** | 21.10 | 22.10 | 14.80 | 15.20 | **18.30** | 19.20 | 22.30 | 13.50 | 14.00 | **17.25** |
| Oxygen % | surface | 74.90 | 87.20 | 86.00 | 83.90 | **83.00** | 100.10 | 98.30 | 92.30 | 100.80 | **97.88** | 91.60 | 102.50 | 75.50 | 92.70 | **90.58** |
| | bottom | 90.20 | 90.00 | 83.10 | 82.00 | **86.33** | 115.40 | 111.50 | 101.90 | 110.10 | **109.73** | 126.20 | 112.70 | 82.70 | 96.70 | **104.58** |
| Sal. ‰ | surface | 38.50 | 38.50 | 38.00 | 38.30 | **38.33** | 37.20 | 37.50 | 38.70 | 38.40 | **37.95** | 37.20 | 37.30 | 38.80 | 38.20 | **37.88** |
| | bottom | 38.60 | 38.50 | 38.30 | 38.50 | **38.48** | 37.90 | 37.60 | 38.80 | 38.90 | **38.30** | 37.90 | 37.80 | 38.90 | 38.80 | **38.35** |
| **Station MG**<br>Water column, 25 m | lat. 40°27′35″ N, long. 17°13′28″ E | | | | | | | | | | | | | | | |
| Sample vol., m$^3$ | | 3.30 | 3.30 | 3.00 | 3.10 | **3.18** | 2.80 | 3.10 | 2.60 | 2.80 | **2.83** | 3.10 | 3.00 | 3.50 | 2.60 | **3.05** |
| Temp. °C | surface | 25.00 | 20.30 | 13.10 | 14.90 | **18.33** | 27.30 | 19.40 | 13.60 | 14.80 | **18.78** | 26.70 | 26.80 | 12.10 | 13.30 | **19.73** |
| | bottom | 23.00 | 15.10 | 12.80 | 14.00 | **16.23** | 22.40 | 19.20 | 14.20 | 14.60 | **17.60** | 23.80 | 25.70 | 12.90 | 14.00 | **19.10** |
| Oxygen % | surface | 86.70 | 82.00 | 84.40 | 85.00 | **84.53** | 110.50 | 102.70 | 100.00 | 103.00 | **104.05** | 95.20 | 97.00 | 88.80 | 67.80 | **87.20** |
| | bottom | 89.30 | 88.60 | 87.90 | 86.70 | **88.13** | 101.90 | 92.70 | 102.20 | 104.50 | **100.33** | 108.10 | 100.20 | 92.40 | 79.40 | **95.03** |
| Sal. ‰ | surface | 39.00 | 37.80 | 37.80 | 38.20 | **38.20** | 38.30 | 38.00 | 37.80 | 37.40 | **37.88** | 37.20 | 37.40 | 38.10 | 38.10 | **37.70** |
| | bottom | 38.60 | 38.60 | 38.00 | 38.50 | **38.43** | 37.70 | 38.50 | 38.60 | 38.30 | **38.28** | 37.70 | 37.80 | 38.80 | 38.70 | **38.25** |

**Table 1.** *Cont.*

| | | 1990–1991 | | | | | 2005–2006 | | | | | 2020–2021 | | | | |
|---|---|---|---|---|---|---|---|---|---|---|---|---|---|---|---|---|
| | | SUM (Jul.) | AUT (Oct.) | WIN (Jan.) | SPR (Apr.) | Average 1990–1991 | SUM (Jul.) | AUT (Oct.) | WIN (Jan.) | SPR (Apr.) | Average 2005–2006 | SUM (Jul.) | AUT (Oct.) | WIN (Jan.) | SPR (Apr.) | Average 2020–2021 |
| **Station MP I** Water column, 12 m | lat. 40°29′24″ N, long. 17°15′54″ E | | | | | | | | | | | | | | | |
| Sample vol., m$^3$ | | 1.30 | 1.50 | 1.10 | 1.20 | **1.28** | 1.60 | 1.40 | 1.70 | 1.50 | **1.55** | 1.20 | 2.10 | 3.00 | 1.60 | **1.98** |
| Temp. °C | surface | 25.60 | 21.10 | 11.90 | 15.20 | **18.45** | 28.10 | 19.40 | 12.40 | 14.80 | **18.68** | 27.30 | 27.50 | 10.60 | 13.00 | **19.60** |
| | bottom | 24.00 | 21.80 | 13.20 | 14.50 | **18.38** | 26.00 | 19.80 | 13.90 | 14.80 | **18.63** | 25.40 | 26.60 | 12.10 | 13.40 | **19.38** |
| Oxygen % | surface | 90.00 | 84.30 | 90.30 | 90.10 | **88.68** | 97.70 | 99.10 | 99.70 | 103.90 | **100.10** | 98.40 | 101.70 | 86.30 | 74.30 | **90.18** |
| | bottom | 80.10 | 86.80 | 89.00 | 88.00 | **85.98** | 84.00 | 88.20 | 97.80 | 109.70 | **94.93** | 104.20 | 104.80 | 94.30 | 87.00 | **97.58** |
| Sal. ‰ | surface | 37.60 | 37.70 | 36.30 | 36.50 | **37.03** | 37.10 | 37.20 | 36.10 | 35.60 | **36.50** | 35.80 | 35.80 | 36.70 | 36.60 | **36.23** |
| | bottom | 38.30 | 38.10 | 37.60 | 38.10 | **38.03** | 38.60 | 37.9 | 37.10 | 38.40 | **38.03** | 37.40 | 37.30 | 38.40 | 38.20 | **37.83** |
| **Station MP II** Water column, 9 m | lat. 40°29′28″ N, long. 17°18′00″ E | | | | | | | | | | | | | | | |
| Sample vol., m$^3$ | | 1.40 | 1.20 | 1.00 | 1.10 | **1.18** | 1.30 | 1.10 | 1.20 | 1.40 | **1.25** | 1.10 | 0.40 | 1.20 | 1.50 | **1.05** |
| Temp. °C | surface | 25.90 | 20.30 | 10.20 | 15.30 | **17.93** | 28.90 | 21.00 | 11.20 | 15.00 | **19.03** | 27.90 | 27.50 | 10.00 | 12.60 | **19.50** |
| | bottom | 23.40 | 20.20 | 11.00 | 15.10 | **17.43** | 27.10 | 21.40 | 12.50 | 14.70 | **18.93** | 26,20 | 26.80 | 11.00 | 13.60 | **19.40** |
| Oxygen % | surface | 88.60 | 87.00 | 87.10 | 87.60 | **87.58** | 101.00 | 99.80 | 117.60 | 106.20 | **106.15** | 99,20 | 95.20 | 91.00 | 78.30 | **90.93** |
| | bottom | 84.50 | 82.70 | 88.00 | 88.00 | **85.80** | 91.30 | 96.10 | 107.20 | 110.30 | **101.23** | 104.60 | 90.30 | 97.00 | 90.00 | **95.48** |
| Sal. ‰ | surface | 37.70 | 37.50 | 36.00 | 36.50 | **36.93** | 37.00 | 37.20 | 35.90 | 36.10 | **36.55** | 35.80 | 36.10 | 35.90 | 36.30 | **36.03** |
| | bottom | 38.20 | 37.60 | 37.10 | 37.50 | **37.60** | 37.90 | 37.60 | 37.90 | 37.90 | **37.83** | 36.70 | 37.00 | 36.50 | 37.30 | **36.88** |

A total of 69 species were identified in the whole set of samples (three time points, four stations, and four seasons) (Table 2). The bulk of the specimens, however, belonged to only 12 taxa representing more than 97% of the total individuals counted, with *Oithona nana* being predominant. A total of 53, 52, and 51 species were found in periods T1, T2, and T3, respectively (Table 2). A general tendency towards an increase in species richness was observed from the external station (G, 61 species) to the internal station (MPII, 25 species) in the period analyzed (Table 2). On the contrary, average specimen abundance per sample increased from G (1493 ind. m$^{-3}$) to MPII (10,926 ind. m$^{-3}$). This last result was also evident in each season. The Spring samples had the lowest species richness (53 species), while the other seasons were equally rich (61, 60, and 61 species for SUM, AUT, and WIN, respectively).

**Table 2.** Copepoda *taxa* distribution in three periods (T1: 1990–1991; T2: 2005–2006; T3: 2020–2021), four stations (G, MG, MPI, MPII), and four Seasons (SUMer, AUTumn, WINter, SPRing). Numbers indicate the average species abundance (ind. m$^{-3}$) for one sample collected at each of the Time, Stations, and Seasons considered. In the table, Copepoda species are listed in alphabetic order of Family and separated as Calanoida (51 species), Cyclopoida (11 species), Harpacticoida (6 species), and Monstrilloida (1 species).

| | **Total, Times** | | | **Total, Stations** | | | | **Total, Seasons** | | | |
|---|---|---|---|---|---|---|---|---|---|---|---|
| **TAXON** | **T1** | **T2** | **T3** | **G** | **MG** | **MPI** | **MPII** | **SUM** | **AUT** | **WIN** | **SPR** |
| *Acartia adriatica* | | 0.06 | | | 0.17 | | | | 0.17 | 0.08 | |
| *Acartia clausi* | 305.16 | 100.10 | 84.95 | 164.27 | 412.18 | 222.27 | 395.24 | 164.27 | 599.08 | 356.31 | 834.36 |
| *Acartia discaudata var. medit* | 0.35 | 2.72 | 0.38 | 1.29 | 6.49 | 0.92 | | 1.29 | 7.00 | 4.54 | 0.70 |
| *Acartia italica* | 48.00 | 21.80 | 12.84 | | 83.17 | 56.36 | 63.73 | | 116.19 | 86.58 | 127.79 |
| *Acartia margalefi* | 0.19 | 3.19 | 0.10 | 0.51 | 2.75 | 1.02 | 4.87 | 0.51 | 2.89 | 5.27 | 5.26 |
| *Acartia negligens* | 0.20 | 1.24 | 1.51 | 4.16 | 1.71 | | | 4.16 | 3.73 | 3.67 | 0.27 |
| *Acartia tonsa* | | | 0.25 | | 0.28 | 0.06 | | | 0.62 | 0.34 | |
| *Paracartia latisetosa* | 81.38 | 6.64 | 5.91 | | 71.33 | 138.97 | 31.70 | | 80.90 | 154.02 | 140.20 |
| *Pteriacartia josephinae* | 2.76 | 1.19 | 1.18 | 0.62 | 11.00 | | | 0.62 | 14.15 | 1.58 | 3.69 |
| *Augaptilus* sp. | 0.11 | 0.21 | 0.11 | 1.01 | | | | 1.01 | 0.15 | 0.28 | 0.30 |
| *Haloptilus longicornis* | | 0.20 | | 0.48 | | | | 0.48 | | 0.27 | |
| *Calanopia* sp. | 0.04 | | | 0.10 | | | | 0.10 | | | 0.05 |
| *Calanus helgolandicus* | | 6.94 | | 12.42 | 6.08 | | | 12.42 | 6.08 | 9.25 | |
| *Mesocalanus* sp. | | 0.05 | | 0.14 | 0.00 | | | 0.14 | | 0.07 | |
| *Nannocalanus minor* | 0.25 | 1.21 | 0.59 | 4.01 | 0.67 | | | 4.01 | 1.46 | 2.40 | 0.33 |
| *Neocalanus* sp. | | 0.03 | 0.29 | 0.08 | 0.38 | | | 0.08 | 0.77 | 0.43 | |
| *Candacia undet.* | 1.18 | 1.49 | 1.30 | 8.27 | 0.58 | | | 8.27 | 2.88 | 2.53 | 1.80 |
| *Paracandacia* sp. | 0.24 | | | 0.63 | | | | 0.63 | | | 0.32 |
| *Centropages kroyeri* | 29.69 | 5.90 | 5.33 | 2.81 | 64.81 | 32.10 | 2.75 | 2.81 | 77.13 | 41.70 | 42.34 |
| *Centropages hamatus* | | | 0.20 | | 0.27 | | | | 0.54 | | |
| *Centropages ponticus* | 33.69 | 0.42 | 23.93 | 1.38 | 63.90 | 51.18 | 6.40 | 1.38 | 121.77 | 53.92 | 52.49 |
| *Centropages typicus* | 11.43 | 5.51 | 8.92 | 24.47 | 28.96 | 0.53 | 2.75 | 24.47 | 40.92 | 8.26 | 21.32 |
| *Centropages violaceus* | 1.53 | 24.30 | | 0.19 | 52.17 | 11.50 | 5.00 | 0.19 | 52.17 | 43.90 | 7.04 |
| *Isias clavipes* | 57.46 | 175.08 | 162.92 | 55.97 | 622.27 | 117.03 | 42.06 | 55.97 | 1029.05 | 375.71 | 120.71 |
| *Clausocalanus arcuicornis* | 12.76 | 4.69 | 0.99 | 22.01 | 24.91 | 0.92 | | 22.01 | 26.52 | 7.17 | 17.55 |
| *Clausocalanus furcatus* | 19.26 | 3.32 | 36.81 | 36.14 | 71.90 | 1.25 | | 36.14 | 121.02 | 50.51 | 26.03 |
| *Clausocalanus joboei* | 4.91 | | 6.26 | 18.17 | 3.03 | | 0.25 | 18.17 | 11.38 | 2.15 | 7.27 |
| *Clausocalanus paululus* | 3.26 | | 2.08 | 9.89 | 1.58 | | | 9.89 | 4.82 | | 4.56 |
| *Calocalanus* sp. | 15.20 | 45.27 | 5.84 | 113.55 | 28.58 | 1.89 | 25.00 | 113.55 | 36.70 | 66.47 | 46.62 |
| *Paracalanus* spp. | 85.09 | 1001.82 | 286.98 | 83.14 | 2378.33 | 634.82 | 179.94 | 83.14 | 3023.43 | 2003.70 | 350.83 |
| *Parvocalanus* sp. | 0.06 | | 0.08 | 0.16 | | 0.11 | | 0.16 | 0.11 | 0.11 | 0.19 |
| *Pseudocalanus* sp. | 0.40 | 1.00 | | 1.90 | 1.67 | 0.17 | | 1.90 | 1.67 | 1.50 | 0.54 |

**Table 2.** *Cont.*

| TAXON | Total, Times | | | Total, Stations | | | | | Total, Seasons | | |
|---|---|---|---|---|---|---|---|---|---|---|---|
| | T1 | T2 | T3 | G | MG | MPI | MPII | SUM | AUT | WIN | SPR |
| *Ctenocalanus vanus* | 1.29 | 4.83 | 15.15 | 28.90 | 7.54 | 0.08 | | 28.90 | 27.78 | 7.49 | 2.39 |
| *Diaixis* sp. | 0.41 | 1.74 | | 4.51 | 1.23 | | | 4.51 | 1.23 | 2.33 | 0.54 |
| *Eucalanus* sp. | 11.53 | 0.11 | 0.78 | 3.29 | 23.78 | 5.00 | | 3.29 | 25.53 | 5.14 | 15.37 |
| *Pareuchaeta* sp. | 0.26 | 2.41 | | 6.77 | 0.33 | | | 6.77 | 0.33 | 3.21 | 0.34 |
| *Heterorhabdus papilliger* | | 0.56 | | 1.45 | 0.05 | | | 1.45 | 0.05 | 0.75 | |
| *Lucicutia sp.* | 0.69 | 0.48 | 1.83 | 4.72 | 0.74 | | | 4.72 | 3.39 | 1.18 | 2.20 |
| *Macandrewella* sp. | 0.31 | | 0.62 | 0.38 | 1.29 | | | 0.38 | 2.12 | 0.83 | 0.42 |
| *Scolecithricella* sp. | 0.38 | | | 1.02 | 0.00 | | | 1.02 | | | 0.51 |
| *Scolecithrix* sp. | 0.38 | | 0.66 | 1.80 | 0.11 | | | 1.80 | 0.99 | 0.17 | 0.54 |
| *Mecynocera clausi* | 0.77 | 5.93 | 1.21 | 15.12 | 3.86 | 0.17 | 0.33 | 15.12 | 5.52 | 9.35 | 1.47 |
| *Pleuromamma gracilis* | 0.10 | 0.32 | | 1.37 | | | | 1.37 | | 0.43 | 0.13 |
| *Pleuromamma xiphias* | 0.00 | 0.16 | 0.70 | 1.35 | | | | 1.35 | 0.94 | 0.42 | |
| *Metridia* sp. | 0.61 | 0.18 | 0.19 | 2.12 | | | | 2.12 | 0.26 | 0.24 | 0.81 |
| *Anomalocera patersoni* | | 0.06 | | 0.17 | | | | 0.17 | | 0.08 | |
| *Pontella* sp. | 0.14 | | 2.35 | 3.00 | 0.33 | | 0.17 | 3.00 | 3.47 | | 0.35 |
| *Pseudodiaptomus* sp. | | | 1.00 | | 1.33 | | | | 2.65 | 1.33 | |
| *Temora longicornis* | | 4.31 | 1.01 | 1.83 | 9.67 | 1.34 | | 1.83 | 11.29 | 8.16 | |
| *Temora stylifera* | 11.11 | 3.83 | 39.57 | 39.22 | 47.34 | 5.22 | | 39.22 | 108.47 | 54.58 | 14.94 |
| *Temora turbinata* | 1.73 | | | 2.88 | 1.75 | | | 2.88 | 1.75 | | 2.31 |
| *Agetus* sp. | | 0.36 | | 0.89 | | | | 0.89 | | 0.48 | |
| *Corycaeus* spp. | 0.06 | | 11.52 | 10.34 | 3.33 | 1.69 | 0.17 | 10.34 | 23.73 | 12.03 | 0.25 |
| *Farranula rostrata* | 5.66 | 5.81 | 7.64 | 24.00 | 15.15 | 1.44 | 0.20 | 24.00 | 32.37 | 11.49 | 7.90 |
| *Urocorycaeus* sp. | 0.03 | | 0.94 | 0.49 | 0.33 | 0.51 | | 0.49 | 2.43 | 0.69 | 0.08 |
| *Oithona nana* | 743.97 | 1580.46 | 1864.63 | 590.10 | 2494.86 | 9059.06 | 30,921.56 | 590.10 | 6110.44 | 12,400.89 | 32,033.11 |
| *Oithona plumifera* | | 12.32 | 0.09 | 24.37 | 8.61 | | | 24.37 | 8.86 | 16.43 | |
| *Oithona similis* | 72.48 | 78.29 | 48.88 | 209.22 | 219.07 | 38.74 | 0.20 | 209.22 | 307.52 | 175.79 | 98.22 |
| *Triconia conifera* | | 0.67 | | 1.61 | 0.17 | | | 1.61 | 0.17 | 0.89 | |
| *Oncaea* spp. | 156.93 | 951.01 | 222.53 | 2323.74 | 850.27 | 100.79 | 45.93 | 2323.74 | 1287.85 | 1409.44 | 303.02 |
| *Copilia* sp. | 0.07 | 0.03 | 0.07 | 0.36 | | | | 0.36 | 0.09 | 0.13 | 0.09 |
| *Vettoria* sp. | | 0.15 | | 0.39 | | | | 0.39 | | 0.19 | |
| *Canuella* sp. | 50.19 | 14.47 | 20.56 | | 14.55 | 18.63 | 160.49 | | 50.28 | 50.35 | 227.41 |
| *Clytemnestra rostrata* | 0.31 | 0.68 | 0.06 | 0.78 | 1.33 | 0.58 | 0.00 | 0.78 | 1.41 | 1.49 | 0.49 |
| *Euterpina acutifrons* | 466.09 | 381.71 | 447.47 | 249.21 | 1551.61 | 1379.52 | 462.63 | 249.21 | 2316.31 | 2294.45 | 1099.68 |
| *Microsetella* spp. | 47.36 | 113.87 | 47.98 | 222.62 | 217.17 | 46.91 | 7.21 | 222.62 | 320.38 | 206.35 | 78.53 |
| *Macrosetella gracilis* | 0.23 | | 0.23 | 0.92 | | | | 0.92 | 0.31 | | 0.61 |
| *Tisbe* sp. | 0.31 | | 5.47 | | | | 8.12 | | 7.29 | 7.29 | 8.54 |
| *Monstrilla* sp. | | 0.58 | 0.16 | 0.03 | 1.08 | 0.60 | 0.05 | 0.03 | 1.30 | 1.37 | 0.27 |
| Totals | 2330.81 | 4948.85 | 3694.89 | 1493.25 | 3308.62 | 4124.63 | 10,926.87 | 1493.25 | 5692.86 | 7024.08 | 12,072.73 |

Three species (*Acartia tonsa*, *Centropages hamatus*, and *Pseudodiaptomus* sp.) were not present in samples from the past (T1 and T2) but appeared in the system at T3. On the contrary, 18 species reported from T1 and/or T2 were not found at T3. *Acartia tonsa* and *Centropages hamatus* were new species in the Taranto area, and *Pseudodiaptomus* sp. is probably a species new to the Mediterranean [19]. Taxonomic re-analysis of samples from the past allowed the identification of *Temora turbinata,* a species never previously reported in the Italian fauna [20], in T1, but not found in samples from the successive periods T2 and T3.

Apart from the 18 species registered in the past collections and absent in the T3 collection, and the dominance of *Oithona nana* in all the Ts, other species dominant in T1 (*Acartia clausi*, *Euterpina acutifrons*) were substituted in T2 (by *Paracalanus* sp., *Oncaea* sp.), and in T3 (by *E. acutifrons*, *Paracalanus* sp.).

Station MPI, unlike the other stations, showed a clear increase in species richness in 2020–2021 (T3), with 27 species, against 21 and 19 species from T1 and T2, respectively. This increase is the result of species richness in the Summer of 2020, with 16 species (against 10 and 11 species from the same season in T1 and T2, respectively).

A total of 35 species were present at all time points, while 16 were linked to a single time point. A total of 18 species were present in all 4 stations, while 16 species were linked to a single station. A total of 39 species were present in all the seasons, while only 1 was linked to a single season (Table 2).

The ANOSIM procedure carried out on samples from the three periods (T) confirmed that each station is significantly different from the others (R = 0.462; $p$ = 0.001). In detail, the difference between sites G and MPII was higher (dissimilarity 71.67%) at the two extremities of the examined system. The lowest dissimilarity (48.50%) was between G and MG (Figure 2). Similar results were observed at each T.

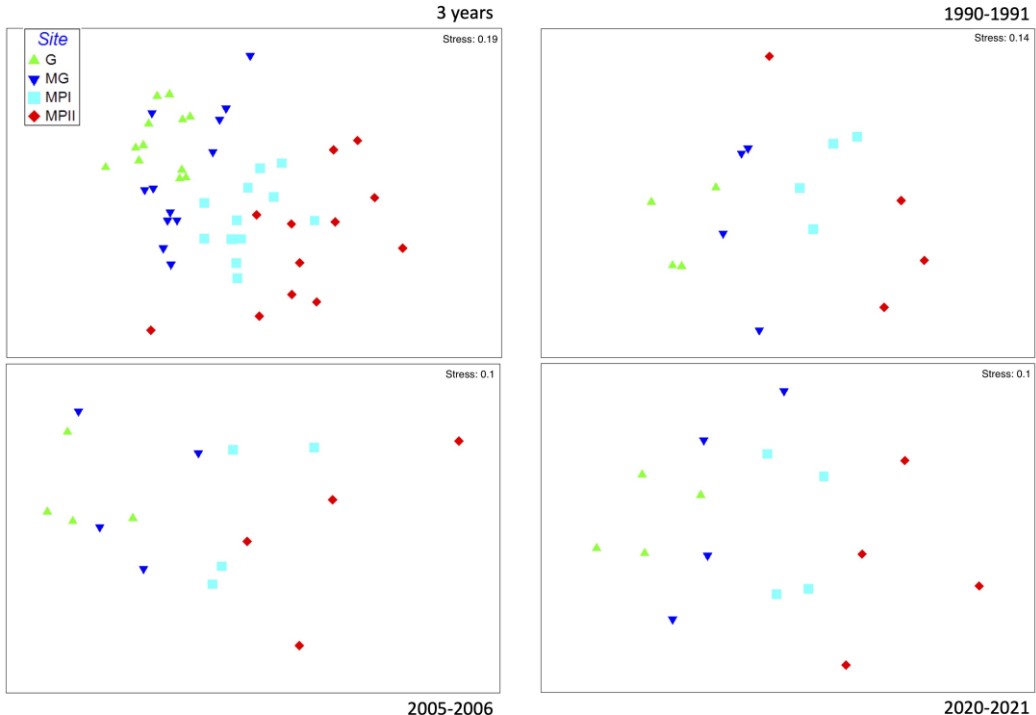

**Figure 2.** nMDS of the zooplankton distribution along the confinement gradient of the Taranto sea system. ANOSIM was carried out on a matrix of 69 taxa × 48 dates/sites.

Regarding individual stations, SIMPER analysis identified station G as the most stable (lowest variation in species composition and specimen abundance) and station MPII as the most variable (overall in terms of specimen abundance).

Regarding the comparison between different Ts, statistical analysis showed that the differences in community composition and abundance measured in the three periods were not significant.

## 4. Discussion

The confined parts of coastal sites are subject to more environmental variations than the open sea areas and are generally exposed to higher human pressures in terms of enhanced eutrophication and pollutant accumulation. The present study was not planned for a detailed study of abiotic conditions, hence the parameters measured were not enough to carry out a robust comparison between past climatic conditions and the environmental changes in the Taranto seas over the last 30 years. The increasing instability in annual environmental conditions from the Gulf to Mar Piccolo II inlet, however, was a constant feature of the growing confinement verifiable from Gulf to Mar Piccolo in the Taranto

sea system. The biotic fraction considered was formed by the planktonic Copepoda, which are the main zooplankton components in this system [1], similar to many other coastal marine areas of the Mediterranean sea [21–26]. Zooplankton in confined areas are generally different from those in the open sea due to a simplification in the community composition, and a reduction in the body sizes of adults [2] of select species highly tolerant and/or able to rest in unfavorable conditions without abandoning the site [27]. The Copepoda assemblage found in the present study was typically richer in species in the open-neritic area and poorer in the innermost part where only the species more tolerant can thrive. The growing importance of *Oithona nana* (from T1 to T3) and the dominance substitution of *Acartia clausi* (T1) with smaller species (*Paracalanus* sp., *Oncaea* sspp., and *Euterpina acutifrons*) in successive T2 and T3 periods depicted a situation in agreement with the general climate warming [2] and/or unstable increase at the innermost stations. Additionally, the occurrence of new species is in agreement with such a changing scenario. Not considering rare species (whose presence/absence could be the result of the sampling process), *Acartia tonsa* was abundant in MG and MPI during T3. The species has been present in the Mediterranean sea since 1985 [28] and shows an interesting enlargement of its geographic distribution in the southward direction. The presence of *Pseudodiaptomus* sp., although rare, is an interesting report because it represents a genus reported as NIS in the Mediterranean Sea in the last 20 years [29]. The present *Pseudodiaptomus* sp. is not *P. marinus* already reported as a NIS in the Mediterranean, and its identification will be the object of a separate dedicated study. *A. tonsa* is a species from northern Europe [30] while *Pseudodiaptomus* sp. is from the Indo-Pacific [19]. The identification of non-indigenous species in the Taranto zooplankton is additional proof of the instability in the community in the present period.

The cited scheme, however, was neglected in the period immediately following the lockdown of human activities from March to May 2020 after the COVID-19 pandemic. In the Mar Piccolo I inlet, we detected an unusual richness in species in the samples collected in the Summer of 2020, a situation typical of less confined sites, and which was not registered in corresponding seasons in T1 and T2, 15 and 30 years before. Recently, a study [26] established an increase in temperature and marinization in the lagoon of Venice, which seems to have driven the new organization in the lagoon zooplankton, including the arrival of new species. The ongoing climate change appears, in that study, to parallel the human disturbance/pressure in the final results (smaller species and lower species richness).

An interesting opposite trend of lowering human pressures on the environment (in terms of concentration of pollutants in the air and water) was reported during the 2020 COVID-19 lockdown [7]. In the North Adriatic area, a similar opposite trend in a biologic indicator (Chl-*a* concentration) was also recorded [9], suggesting that the interruption of human activity, although short, could be responsible for prompt rehabilitation of impacted environments.

Our results are consistent with the general trend of climate–environment evolution, either with regards to the dominance of small species (*Oithona nana*, *Paracalanus* sspp., *Oncaea* sspp., and *Euterpina acutifrons*) and/or the arrival of new species. *Pseudodiaptomus* sp. is a congeneric of a species identified in Venice (*P. marinus*), testifying to the facilitation of such new arrivals amidst the ongoing climate change, as well as the arrival and identification of a new genus in the Mediterranean sea. Apart from this trend, which is in accordance with the literature, we documented an original situation where the COVID-19 lockdown appeared to have removed, just for a short time, the component of human disturbance on the system, thus allowing its evolution towards a situation typical of less confined areas (more species).

The MPI area is the most affected by human settlement and the absence of any movement of boats is an unusual situation that MPI experienced in the post-COVID-19 lockdown period. This result of high species richness did not affect successive seasons, probably because the re-opening of the free circulation re-established the pre-COVID-19 level of disturbance. Other stations (G, MG, MPII) did not show an increase in species numbers,

thus suggesting that MPI is the most stressed area—due to human pressure—among those studied.

## 5. Conclusions

The general trend of climate warming produces alterations in structure and composition of zooplankton in confined coastal environments. Such an effect is synergistic with human pressure on the environment, which generally increases with confinement. The zooplankton in the Taranto sea (a system with different degrees of confinement) appear to follow this rule over long periods, based on comparisons of species assemblage with those from 15 and 30 years ago. The number of species remained constant in the three compared periods, with species with small bodies increasing in importance and new species arriving in the system substituting older ones.

The present study, however, showed a short-lasting (only one date of sampling) and space-limited (only station MPI) recovery of the system, with more species, after human activities were stopped with the imposition of lockdown during the COVID-19 pandemic. The narrow localization of the recovery, and its short duration, however, require additional elements and data for correct interpretation. This notwithstanding, the short recovery is an encouraging effect of the reduction in human pressure on the environment, suggesting that this could be a localized solution for mitigating or interrupting the ongoing changes due to climate warming.

**Author Contributions:** Conceptualization, G.B. and F.R.; funding acquisition, G.B. and F.R.; methodology, G.B.; supervision, G.B.; writing, F.R.; formal analysis, G.D.; visualization, G.D.; Project administration, F.R. All authors have read and agreed to the published version of the manuscript.

**Funding:** This project was partly funded under the National Recovery and Resilience Plan (NRRP), Mission 4 Component 2 Investment 1.4—Call for tender No. 3138 of 16 December 2021, rectified by Decree n.3175 of 18 December 2021 of the Italian Ministry of University and Research funded by the European Union—NextGenerationEU. Award number: Project code CN_00000033, Concession Decree No. 1034 of 17 June 2022 adopted by the Italian Ministry of University and Research, Project title "National Biodiversity Future Center—NBFC". Part of the funds derive also by Project SNAP-SHOT (Synoptic Assessment of Human Pressures on Key Mediterranean Hot Spots), funded by the Department "Sciences of the Earth System and Technologies for the Environment" of the National Research Council of Italy.

**Informed Consent Statement:** Not applicable.

**Data Availability Statement:** Not applicable.

**Acknowledgments:** The Marine Biology Museum "P.Parenzan", University of Salento, stores samples and species highlighted by the present study.

**Conflicts of Interest:** The authors declare no conflict of interest.

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
