# Peer review of "Planktonic Copepoda along the Confinement Gradient of the Taranto Sea System (Southern Italy) after Lockdown of Human Activities Due to the COVID-19 Pandemic"

_water, doi:10.3390/w15132449_

Round 1

Reviewer 1 Report

Review for the paper "Copepoda along the confinement gradient of the Taranto sea system (southern Italy), after the lockdown of human activities due to COVID 2019 pandemic event" by Belmonte Genuario, Denti Giuseppe, Rubino Fernando submitted to "Water".

General comment:

 Planktonic copepods in the Mediterranean Sea have been extensively studied; however, the current distribution and abundance patterns in the coastal waters of the Taranto Sea remain incompletely understood. The COVID-19 pandemic has significantly impacted all human activities worldwide, including coastal management, exploitation of biotic resources, and industries operating in inshore waters. As Copepoda is one of the most sensitive groups to environmental changes, it can serve as a reliable indicator of shifts related to human activity. The authors conducted research to determine whether the pandemic might have influenced copepod populations. They discovered increased species richness in the summer of 2020 compared to previous years, attributing this to reduced ship and boat traffic in the area. The study relies on four seasonal samplings, with valid and appropriately applied data collection and processing methods. Results are presented through relevant figures and tables. My primary concern lies in the interpretation of the main findings. The authors focused solely on human activity and did not adequately consider environmental factors. Consequently, their principal conclusion appears speculative. I suggest a major revision centered primarily on the role of environmental factors. The potential significance of the COVID-19 pandemic should be mentioned as a secondary factor that may influence zooplankton patterns in the study area.

 Major concerns.

 1. I recommend reevaluating the study's aim, placing greater emphasis on exploring the current spatial patterns in copepod assemblages and comparing them to previous findings. Given the numerous copepod studies conducted in the region in recent years, the novelty of the present study must be more clearly emphasized.

 2. Performing statistical comparisons across different years and seasons would aid in better understanding the inter-annual changes in copepod community structure.

 3. Discussion: The study's results should be compared with data from other similar locations to provide a comprehensive analysis.

 4. Discussion: The interpretation of the results must primarily focus on natural factors, such as climate, hydrology, currents, water masses, predator-prey interactions, and food availability. In particular, the initiation date and duration of the spring phytoplankton bloom can directly impact zooplankton composition and abundance. The intrusion of deep waters and advection of water masses may be responsible for seasonal and year-to-year variations in copepod assemblages. Additionally, effects from larval fish and jellyfish cannot be disregarded. While I have mentioned only the most obvious and significant factors, many others should be detailed further. Human impacts ought to be considered as secondary factors in the analysis.

 Specific remarks.

 L24. Consider replacing "The taxon Copepoda was individuated as representant" with "Copepods were selected as the best indicators".

 L107. Indicate the mouth area of the net.

L169-173: I recommend comparing environmental variables between seasons and years using appropriate statistical methods such as one-way ANOVA or Kruskal-Wallis tests.

 L184 and below: It is advisable to round off copepod abundance values to the nearest whole numbers (e.g., 1,493 ind. m³ instead of 1,493.25 ind. m³).

 L209-220: It is crucial to conduct relevant statistical comparisons of zooplankton abundance in varying years (seasons) in order to demonstrate the significance of inter-annual/seasonal differences.

 Fig. 2 needs improvement, as it is currently not easily visible. It is recommended to increase the font size for better readability.

 L261: The species name, P. marinus, should be italicized in the text.

The English should be carefully revised.

Author Response

We hope to have answered to all the questions satisfactorily.

point by point response is provided at the end of the present message (responses in bold).

all the changes (also those suggested by other reviewers) now appear as red text in the manuscript (word file)

thank you for the discussion and the attention.

Manuscript ID: water-2452284
Type of manuscript: Article
Title: Copepoda along the confinement gradient of the Taranto sea system (southern Italy), after the lockdown of human activities due to COVID 2019 pandemic event
Authors: Genuario Belmonte *, Fernando Rubino, Giuseppe Denti

Submitted to section: Biodiversity and Functionality of Aquatic Ecosystems,
special issue “Implementation of Biodiversity and Ecosystem Services in Marine Ecosystem
Management”, Volume II

Reviewer I

My primary concern lies in the interpretation of the main findings. The authors focused solely on human activity and did not adequately consider environmental factors. Consequently, their principal conclusion appears speculative. I suggest a major revision centered primarily on the role of environmental factors. The potential significance of the COVID-19 pandemic should be mentioned as a secondary factor that may influence zooplankton patterns in the study area.

The environmental parameters which have been considered (T, Sal, D.Oxygen) have not the right number of data/values (only four, per year) to sustain a robust statistic on the environment role on the community change. In fact, the statistical tests did not give support to such a hypothesis. The tendence, however, is that of a general warming of the system (as in any other part of the world) but this does not agree with an increase of the species richness. What has been found, in fact, is a result not justified by the warming of the system, and the explanation is that of a human activity suspension just because the increase has been registered immediately after the Covid lockdown.

Major concerns.

  1. I recommend reevaluating the study's aim, placing greater emphasis on exploring the current spatial patterns in copepod assemblages and comparing them to previous findings. Given the numerous copepod studies conducted in the region in recent years, the novelty of the present study must be more clearly emphasized.

done

  1. Performing statistical comparisons across different years and seasons would aid in better understanding the inter-annual changes in copepod community structure.

This has been performed with ANOSIM and SIMPER. The tables of such results, however, have been omitted to avoid their affection on manuscript length

  1. Discussion: The study's results should be compared with data from other similar locations to provide a comprehensive analysis.

Each estuary or coastal brackish water site is different from the others and comparisons are not reliable. This notwithstanding, general rules obtainable from other studies have been proposed with the addition of a couple of recent references dealing with the COVID lockdown consequences on coastal marine sites.

  1. Discussion: The interpretation of the results must primarily focus on natural factors, such as climate, hydrology, currents, water masses, predator-prey interactions, and food availability. In particular, the initiation date and duration of the spring phytoplankton bloom can directly impact zooplankton composition and abundance. The intrusion of deep waters and advection of water masses may be responsible for seasonal and year-to-year variations in copepod assemblages. Additionally, effects from larval fish and jellyfish cannot be disregarded. While I have mentioned only the most obvious and significant factors, many others should be detailed further. Human impacts ought to be considered as secondary factors in the analysis.

Climate is not sustained by a proper number of recorded values (this is an interpretation, in any cases the considered environmental parameters did not show useful indications for a responsibility of parameters on the finding of more species), hydrology was not considered, water masses movements also, predatory-prey interactions, and food availability were not the target of the proposed manuscript. The consideration of human impact (suspended) is just a suggestion, also derived by the consultation of some recent published papers (see references n. 7, 8, 26 of the manuscript)

 Specific remarks.

L24. Consider replacing "The taxon Copepoda was individuated as representant" with "Copepods were selected as the best indicators".

 done

L107. Indicate the mouth area of the net.

 done

L169-173: I recommend comparing environmental variables between seasons and years using appropriate statistical methods such as one-way ANOVA or Kruskal-Wallis tests.

One-way ANOVA and Krustal-Wallis tests have been used, but results were not statistically significant. Tables have been omitted from the text

L184 and below: It is advisable to round off copepod abundance values to the nearest whole numbers (e.g., 1,493 ind. m⁻³ instead of 1,493.25 ind. m⁻³).

 done

L209-220: It is crucial to conduct relevant statistical comparisons of zooplankton abundance in varying years (seasons) in order to demonstrate the significance of inter-annual/seasonal differences.

Interannual differences were not statistically significant. For this reason we avoided a table with such a result, but now the manuscript contains the explicit declaration of this result (three lines at the end of the result section).

Fig. 2 needs improvement, as it is currently not easily visible. It is recommended to increase the font size for better readability.

done 

L261: The species name, P. marinus, should be italicized in the text.

done

Comments on the Quality of English Language

The English should be carefully revised.

The English language has been checked by Nicolette James a mother language teacher, already familiar with papers on zooplankton.

Reviewer 2 Report

In this study the Authors put in relation the unusual increase of copepod abundance and the arrival of new species in a confined environment to the absence of human disturbance of boat traffic due to the lockdown period of the COVID 19 pandemic event. The MS is interesting, clear, well written and highlight how the human activities can affect some ecosystems more subjected to environmental stress. This study, although restricted in space and time, is a good starting point for future research focalizing to the mitigation of the human disturbance. In my opinion the MS is worthy of publication and only minor changes are required below described.

Table 1: I suggest to highlight the average values (e.g. bold characters or with a grey background).

-Table 2: is very long and difficult to read. It could be move in the supplementary material paragraph and in its place put a graph about the species richness changes in the different stations and in the different sampling periods.

-Table 3: also this table is not very readable…I would substitute it with a summary table reporting only the average dissimilarity values and move the detailed table in the supplementary material paragraph.

Line 261: P. marinus in italic.

Author Response

We thank the reviewer 2 for the suggestions. a word file is annexed with the requested changes (together with those requested by reviewer 1). the point by point answers are here enclosed:

Table 1: I suggest to highlight the average values (e.g. bold characters or with a grey background).

the average values are now in bold 

-Table 2: is very long and difficult to read. It could be move in the supplementary material paragraph and in its place put a graph about the species richness changes in the different stations and in the different sampling periods.

we prefer to maintain table 2 which is the summary of the species distribution in seasons, stations, and periods. the suggestion to delete a table from the text has been followed for table 3

-Table 3: also this table is not very readable…I would substitute it with a summary table reporting only the average dissimilarity values and move the detailed table in the supplementary material paragraph.

table 3 has been deleted from the text. and it is available as supplementary material accordingly the opinion of the editor.

Line 261: P. marinus in italic.

done. OK

Round 2

Reviewer 1 Report

Accept.